# Tocilizumab in Juvenile Idiopathic Arthritis Associated Uveitis, a Narrative Review

**DOI:** 10.3390/children10030434

**Published:** 2023-02-23

**Authors:** Claudia Iannone, Luca Marelli, Stefania Costi, Maria Rosa Pellico, Lamberto La Franca, Roberto Caporali, Elisabetta Miserocchi

**Affiliations:** 1Division of Clinical Rheumatology, ASST Gaetano Pini-CTO Institute, 20122 Milan, Italy; 2Faculty of Medicine, University of Milan, 20122 Milan, Italy; 3Department of Clinical Sciences and Community Health, Research Center for Adult and Pediatric Rheumatic Diseases, University of Milan, 20122 Milan, Italy; 4Ophthalmology Department, Ospedale San Raffaele, University Vita-Salute, 20132 Milan, Italy

**Keywords:** uveitis, tocilizumab, juvenile idiopathic arthritis, bDMARDs

## Abstract

Juvenile idiopathic arthritis (JIA) associated uveitis (JIA-U) is the most common extra-articular manifestation of JIA, affecting 10–15% of patients, especially in oligoarticular JIA where its course may be faint. Therefore, JIA-U is one of the most challenging pediatric uveitis, associated with major ocular morbidity and possibly leading to irreversible structural ocular damage and to vision-threatening complications. Adequate management is crucial for avoiding visual impairment complications. Since the introduction of biologic disease modifying anti-rheumatic drugs (bDMARDS), the visual prognosis of JIA-U has dramatically improved over the decades. Tumor necrosis factor-α (TNF-α) blockers are the most used bDMARDs in treating JIA-U with large evidence of efficacy. However, inadequate response to these agents, either due to intolerance or inefficacy, may be observed, requiring a swap to other classes of immunosuppressive agents, including anti-IL-6, anti-CD20, and, more recently, JAK inhibitors. Tocilizumab is a humanized monoclonal antibody to the interelukin-6 receptor preventing IL-6 from binding to its soluble and membrane-bound receptors. A growing body of literature provides promising results about the efficacy of intravenous and subcutaneous tocilizumab in the treatment of JIA-U. A narrative review of the literature on this topic will improve our knowledge on the potential use of tocilizumab in JIA-U.

## 1. Introduction

Juvenile idiopathic arthritis (JIA) is a heterogeneous group of chronic immune-mediated inflammatory diseases with onset before 16 years of age, persisting for more than 6 weeks, and of unknown etiology. As the name itself may suggest, the main disease targets are the joints [1,2]. However, extra-articular manifestations are often present, uveitis being the most common one [3,4]. 

JIA subset classification is defined according to the type and number of affected joints, autoimmune characteristics (RF positivity), extra-articular manifestations, and genetic factors. Although new classification criteria have been proposed, historically, two main disease subsets are identified according to the number of joints involved: oligoarticular, involving less than five joints, and polyarticular, involving five or more joints [5]. Despite the increasingly effective treatment strategies, the burden of disease in JIA remains high, and assessing disease outcomes becomes challenging when extra-articular manifestations are present. Indeed, articular and ocular clinical responses to therapies may not be concurrent, and the measurement of global disease activity may be partial. 

Uveitis, the inflammation of the uvea comprising the iris, ciliary body, and choroid, is the most frequent extra-articular manifestation of JIA (JIA-U), occurring in up to 20% of children with JIA, almost exclusively in the mono-oligo articular ANA+ subset. JIA-U is the most frequent cause of non-infectious uveitis in childhood, and its typical presentation is of a chronic bilateral anterior uveitis with an insidious onset and asymptomatic course. Up to 38% of patients with JIA-U develop severe visual impairment, and blindness can occur in 16–22% of cases, with cystoid macular edema (CME) being the most common cause of visual loss [3]. 

Furthermore, chronic inflammation causes potentially irreversible ocular complications including cataract, band keratopathy, posterior synechiae, glaucoma, hypotony, and maculopathy. Early identification and prompt aggressive treatment are of pivotal importance to avoid this vision-burdening scenario. Treatment of JIA-U includes the use of topical corticosteroids as well as systemic conventional disease-modifying antirheumatic drugs (DMARDs), mainly methotrexate. Patients who are resistant to first-line topical or systemic therapy are often treated with biologic DMARDs (bDMARDs) [3,4,6]. TNF-alpha inhibitors, anti-CD20, JAK inhibitors, and anti-IL-6 are often administered [7]. 

According to American College of Rheumatology consensus treatment plans, TNF-alpha inhibitors are the preferred choice of treatment for second-line therapy [8] after failure of conventional DMARDs. An alternative second line bDMARD is an IL-6 inhibitor, namely tocilizumab (TCZ).

IL-6, interacting with its membrane bound receptor (IL-6R), plays a pivotal role in driving inflammation and immune response in several diseases. Surface IL-6R is cleaved, and the soluble complex IL-6/IL-6R can activate further cytokine receptors sharing the gp130 molecule; therefore, blocking IL-6R can exert pleiotropic effects beyond the simple inhibition of IL-6 pathway [9]. TCZ is a fully humanized anti-IL-6R antibody approved as therapy for numerous autoimmune diseases, including rheumatoid arthritis (RA), adult-onset Still’s disease, and Castleman disease, as well as for the treatment of systemic JIA, where uveitis is uncommon [10,11]. There is emerging evidence that targeting IL-6R can also be efficacious in treating severe non-infectious uveitis, both idiopathic and associated to JIA. The purpose of this narrative review is to focus on the use of TCZ in JIA-U and to envisage a possible position of TCZ in the therapeutic algorithm of this fearsome clinical manifestation.

## 2. Juvenile Idiopathic Arthritis Associated Uveitis

### 2.1. Epidemiology and Risk Factors

Among extra-articular complications of juvenile idiopathic arthritis (JIA), inflammation of the uvea is the most frequent and severe, possibly leading to permanent loss of vision if untreated [12]. According to the Standardization of Uveitis Nomenclature (SUN) workshop criteria, uveitis as a classification is based on the primary anatomical location (anterior, intermediate, posterior, or panuveitis) and the time-based pattern of inflammation (acute, subacute, chronic, or recurrent), with the chronic anterior type described as prevailing in patients affected by JIA [13,14]. Among young patients with known JIA, the reported prevalence of uveitis ranges from 11.6% to 30%, [3] being higher in Northern (~19%) and Southern Europe (~19%), and lower in Latin America (~6%), Africa, the Middle East (~6%), and Southeast Asia (5%) [15]. The differences reported between geographical regions may reflect significant discrepancies in the availability of ophthalmologic screening between countries. The established risk factors for the occurrence of JIA-associated uveitis, and chronic anterior uveitis in particular, are ANA-positive status, oligoarticular disease, female gender, and early age at onset of diagnosis (≤6 years old) [16,17,18]. 

### 2.2. Pathogenesis

The exact pathophysiology of JIA-U is not fully understood and remains hypothetical, despite the strong association between arthritis and uveitis. Few reports of familial cases of JIA-U suggest a genetic susceptibility, especially with respect to HLA class II genes, but a more complex pathogenesis involving external non-genetic factors is currently proposed, since a low concordance rate among twins has been demonstrated [19,20]. In particular, in children affected by oligoarticular arthritis, JIA-U has been linked with the HLA-DR5 haplotype and HLADRB1*1104 allele [21,22]. HLA-B27 is associated with higher risk of acute anterior uveitis in patients with ERA [23]. HLA-DR1, in contrast, may have a protective role against uveitis onset in these children. Both B- and T-lymphocytes appear to be responsible for the immune response against intraocular antigens such as S-arrestin, retinol-binding protein 3, and tyrosinase-related proteins [22,24]. A crucial role of B-lymphocytes in the process underlying the disease is suggested by transcriptomic and proteomic examinations of iris tissue and aqueous humor in children affected by JIA-U, in which a strong expression of molecules associated with B cells and plasma cells has been revealed. A significantly higher concentration of the B-cell activating and survival factors BAFF, APRIL, and IL-6 in the aqueous humor of JIA-U patients has been registered [25]. Nevertheless, direct T-cell involvement has not been demonstrated in JIA-associated uveitis so far, despite the fact that CD4+ involvement is critical in the pathogenesis of JIA and experimental uveitis [15]. Few studies have investigated the association of external triggers, including infections, vaccines, and environmental factors, and JIA-U, but no causal relationship has been settled to date [19].

### 2.3. Clinical Manifestations and Diagnosis

Acute anterior uveitis is rare in children with JIA and is more common in HLA-B27 positive patients: this type of uveitis usually presents with overt symptoms, such as eye pain, redness, photophobia, and visual changes. Differently, chronic, bilateral, non-granulomatous JIA-U is usually asymptomatic. The diagnosis may be challenging since collecting symptoms and performing examinations are often difficult due to the young age of these patients. Regardless of recent advances in diagnosis and management, 25–50% of patients develop complications, including posterior synechiae, band keratopathy, cataract, glaucoma, and cystoid macular edema, which may influence their quality of life and their ability to accomplish visual tasks and deal with everyday activities [26]. Thus, early detection through regular ophthalmologic screening encompassing slit lamp examination, laser flare-photometry evaluation, dilated fundus oculi evaluation, visual acuity testing, and measurement of intraocular pressure is advisable every three to twelve months, based on each patient’s risk profile [8].

## 3. Treatment of JIA-U

Patients affected by JIA-U should be treated in tertiary care centers where the cooperation between pediatric rheumatology specialists and trained uveitis experts may reduce the rate of sight-threatening complications by improving diagnostic and therapeutic protocols. The target of this interdisciplinary management is to achieve quiescence with zero cells in the anterior chamber (SUN anterior chamber cell grade 0) in both eyes.

### 3.1. Conventional Treatmnent

Topical corticosteroids should be used as first-line treatment to control inflammation, followed by tapering as the anterior chamber cellular reaction comes under control [8]. In order to prevent the formation of posterior synechiae, cycloplegics eye drops, including tropicamide and cyclopentolate, are warranted in association with topical steroids [27]. If topical steroids fail to control the inflammation, oral or intravenous corticosteroids can be used. On the downside, long-term use of both topical and systemic steroids may lead to cataract, especially if more than three drops daily are used [12,28]. Moreover, chronic administration of corticosteroids may raise intraocular pressure, possibly resulting in glaucomatous optic neuropathy and irreversible visual loss [29]. As per the latest Multinational Interdisciplinary Working Group for Uveitis in Childhood (MIWGUC) treatment recommendations, systemic immunosuppression is advised for active JIA-U if there are poor prognostic factors and/or if persistent disease is present, defined as active uveitis lasting >3 months. Men’s sex, ocular complications, and uveitis before arthritis are included in these risk factors [12,30]. Methotrexate (MTX) is the first line of DMARDs, with subcutaneous delivery recommended over the oral one [30,31]. A systematic review published by Simonini et al. has shown that MTX in monotherapy can improve intraocular inflammation in 73% of children affected by refractory autoimmune chronic uveitis [32]. Antimetabolites such as azathioprine (AZA) and mycophenolate mofetil (MMF) seem to be less effective than MTX in controlling JIA-U. Their use, alone or in combination, may be considered in patients’ refractory to other immunosuppressive drugs considering clinician experience, availability of the drugs, and safety profile [33,34].

### 3.2. Biological Treatment and Janus Kinasis Inhibitors 

A considerable number of children affected by JIA-U may not respond to conventional DMARDs or do not tolerate these drugs for long periods. In these cases, it is recommended to add or switch to a bDMARD, according to the SHARE initiative, with TNF-α inhibitors being the first-line therapy [35]. Moreover, in children presenting with sight-threatening complications, starting MTX and a TNF-α inhibitor is conditionally recommended over MTX as monotherapy [8]. Adalimumab, a fully human monoclonal antibody specifically binding to TNF-α, has been approved by the US Food and Drug Administration (FDA) for treatment of non-infectious uveitis in patients older than 2 years of age [36]. Two randomized, placebo-controlled, double-blinded trials have demonstrated its effectiveness in the treatment of patients with JIA-U. In SYCAMORE trial, adalimumab was able to reduce treatment failure rate (27%) compared to the placebo group (60%) in a cohort of patients with active uveitis despite the treatment with MTX. Moreover, a significant steroid-sparing effect was also noticed [37]. In ADJUVITE trial, 31 patients with ocular inflammation refractory to topical steroids and MTX were randomized to receive either adalimumab or placebo. After 60 days of therapy, 56% of young patients in the treatment group achieved a reduction in ocular inflammation as measured by laser flare photometry, compared to 20% in the placebo group. Infliximab is a chimeric monoclonal antibody that blocks TNF-α binding to its receptor. In a prospective, multicenter study conducted on JIA-U patients refractory to standard immunosuppressive treatment and/or corticosteroid-dependent, adalimumab demonstrated a higher remission rate (60.0%) than infliximab (20.3%). Furthermore, the former was generally favored over the latter due to ease of administration with subcutaneous injections as opposed to infusions [38]. Golimumab may represent an effective option, as demonstrated in a retrospective single-center study conducted on ten children with JIA-U refractory to adalimumab, but further evidence is needed to confirm these promising findings [39].

Adalimumab and infliximab are conditionally recommended over etanercept for active JIA-U [8,38]. In fact, while effective for the management of articular symptoms, in a randomized, placebo-controlled, double-masked clinical trial conducted on children with active JIA-associated uveitis, it did not provide a significant efficacy above placebo. Moreover, it may itself induce uveitis in both previously uninvolved and inflamed eyes [40,41]. 

In patients with active JIA-U who have failed MTX and two TNF-α inhibitors, the use of abatacept or TCZ as bDMARD options is recommended [8]. Abatacept is a selective T cell co-stimulation modulator that inhibits the activation of T cell activation by binding to CD80 and CD86 receptors on antigen-presenting cells [42]. Zulian et al. described the use of intravenous abatacept in a prospective study conducted on children affected by JIA-U refractory to both cDMARDs and TNF-α inhibitors. All seven patients responded to abatacept, and 85.7% maintained clinical remission after a mean of 9.2 months of therapy [43]. Despite these results, further evidence is needed to clarify its effectiveness for these patients, since in another study by Tappeiner and colleagues a sustained response to abatacept was not common in a cohort of 21 patients with severe and refractory uveitis, with the latter recrudescing in eight of 11 patients who had reached inactivity and persisting in 10 cases [44].

Rituximab is a chimeric mouse–human monoclonal antibody directed against the CD20 molecule [45]. Even if evidence regarding its efficacy mainly derives from case series, it appears to be a promising option for patients with JIA-U who have not previously responded to first line bDMARDs. Miserocchi et al. reported the complete control of ocular inflammation in eight patients (100%) affected by JIA-U refractory to conventional immunosuppression and TNF-α inhibitors over a mean follow-up of ~45 months; although, in two children, treatment was discontinued because of inefficacy for arthritic disease [46].

Janus kinase inhibitors (JAKi) may represent a new valuable treatment option for children with JIA-U, particularly in those not responding to conventional or biologic DMARDs. Evidence of the efficacy of JAKi in treating these children is still limited. Indeed, one of the pioneering pediatric trials to assess the clinical effectiveness and safety of a JAK inhibitor in JIA-uveitis or chronic ANA-positive uveitis is JUVE-BRIGHT, a multicenter clinical trial that is being conducted on patients who did not achieve an adequate treatment response or showed intolerance to MTX but not bDMARDs. Patients were randomized, and adalimumab was used in the reference arm. However, no result has been published yet, since the trial is still ongoing [47]. Another international, randomized, open-label controlled study on baricitinib use for pediatric uveitis is currently ongoing (ClinicalTrialsGov NCT04088409). In a case series recently published by our group, treatment with baricitinib and tofacitinib was associated with improvement of uveitis without any systemic adverse event. However, the authors observed a different response between the uveitis and the articular disease, with the latter not responding as favorably as the former [48].

## 4. Tocilizumab

IL-6 has been discovered as a cytokine that is promptly and transiently released during the early phase of inflammatory response to infections and tissue injury by activating the inflammatory cascade and driving the immune response. The excess of IL-6 production can induce the so called “cytokine storm” or “hyperinflammation”, which may directly cause tissue damage as in SARS-CoV-2 infection and in the cytokine release syndrome. However, IL-6 has been differently named according to the field of research in which it has been investigated, for instance, B-cell stimulatory factor 2 (BSF-2), due to its ability to promote the differentiation of activated B cells into plasma cells, or hepatocyte-stimulating factor (HSF) for inducing the synthesis of acute phase proteins by hepatocytes, or the (IFN)-b2 owing to its IFN antiviral activity [9]. IL-6 is also involved in regulating bone marrow functions as platelet differentiation or reducing iron absorption through the intestinal epithelia cells or stimulating fibrogenesis by fibroblasts [49]. These pleiotropic functions are directly related to the unique mechanism of signaling of IL-6, the canonical signaling, and the trans-signaling [50] (Figure 1). IL-6 receptor (IL-6R) is expressed on a few cells, such as hepatocytes, fibroblasts, and lymphocytes, as homodimer receptor that is activated upon IL-6 binding and coupling with the signal-transducing receptor β-subunit membrane glycoprotein 130 (gp130), the canonical IL-6 signaling. Other cytokines, such IL-11, IL-27, leukemia inhibitory factor (LIF), and oncostatin M (OSM) can signal through a complex made by its own membrane receptor coupled with gp130. Like many surface receptors, IL-6R is shed from the cell membrane as soluble (s) IL-6R that bind to circulating IL-6 forming sIL6-R/IL-6 capable of activating the prior cytokine receptors containing gp130, the IL-6 trans-signaling. Therefore, this group of molecules is now named IL-6 cytokine family [50]. This premise is important to picture the multiple biological and pathophysiologic roles of IL-6 and to explain why IL-6 is a key factor in such a different disease as rheumatoid arthritis (RA) [51,52], juvenile idiopathic arthritis (JIA) [53], adult-onset Still’s disease (AOSD) [54], giant cell arteritis (GCA) [55] and Castleman disease [56], cytokine release syndrome, and uveitis [19].

Tocilizumab (TCZ), a humanized monoclonal antibody, binds to membrane-bound IL-6 receptor (mIL-6R) and soluble IL-6 receptor (sIL-6R) and inhibits IL-6 signalling by preventing IL-6 from binding to IL-6R. The decision to target IL-6R rather than IL-6 itself was made to make dose and regimen selection easier, since concentrations of the receptor have less interpatient variability when compared to concentrations of IL-6 [5,47]. 

### 4.1. Tocilizumab in JIA

TCZ is employed in numerous inflammatory chronic diseases [50,51]. Since 2005, clinical trials testing tocilizumab in patients with systemic JIA were carried on and led to its employment as a first line drug in this severe pediatric condition, due to its efficacy in ameliorating signs and symptoms of the disease itself, its clinically relevant glucocorticoid-sparing potential, and its ability to reverse growth retardation in these young patients [57,58,59,60,61,62,63,64]. Trials of TCZ were undertaken in polyarticular JIA from 2009 based on results obtained in RA. A strong body of evidence about its efficacy and safety led to its inclusion, as second line agent after failure of MTX treatment, in the 2019 ACR guidelines for the treatment of non-systemic polyarthritis JIA [65]. Targeting IL-6 pathway seems so promising in JIA that another antibody targeting IL-6R, sarilumab, is in phase II trials for polyarticular JIA and systemic JIA (ClinicalTrialsGov NCT02776735; NCT02991469).

### 4.2. Tocilizumab in Non-Infectious Uveitis

Tocilizumab has demonstrated itself to be a powerful weapon also against non-infectious uveitis. In 2014, Papo and colleagues demonstrated the efficacy of TCZ in treating different types of refractory noninfectious uveitis by showing improvement in visual acuity and resolution of CME in six out of eight patients enrolled. Since then, a great deal of other studies has been mushrooming, but their detailed description is beyond the scope of this review. One of the most remarkable among them, both in terms of numbers and variety of included diseases, is the one *published* in 2018 by Sepah et al., which collects 37 patients (*n* = 2 sarcoidosis, *n* = 2 Vogt–Koyanagi–Harada syndrome, *n* = 2 birdshot choroidopathy, *n* = 1 punctate inner choroiditis, *n* = 1 Behçet’s disease, *n* = 1 tubulointerstitial nephritis and uveitis (TINU syndrome), and *n* = 28 idiopathic uveitis). A total of 22 patients were treatment-naïve, and 15 patients had been treated with local or systemic corticosteroids and/or immunomodulatory therapy, which was interrupted at least 30 days before the administration of intravenous TCZ, which was showed to be effective in improving visual acuity and in reducing vitreous haze and central macular thickness. Furthermore, in a multicentric French study, treatment with TCZ was associated independently with complete response of uveitic macular edema, one of the most fearsome manifestations of uveitis, compared with anti-TNF-α agents [66].

### 4.3. Tocilizumab in JIA-U

Tocilizumab has been demonstrated to be a valid therapeutic option in JIA-U. Since 2012, anecdotal reports about the use of TCZ in severe treatment-resistant uveitis came out [67,68,69]. The first pioneering retrospective study concerning this aspect was published in 2016 and assessed the efficacy of TCZ in treating severe and therapy-refractory JIA-U in a cohort of 17 patients with active uveitis. In the whole cohort, uveitis had been resistant to previous topical and systemic steroids, MTX, and other synthetic and biological DMARDs, including ≥1 TNF-α inhibitor. Uveitis inactivity was reached in 10 patients after a mean of 5.7 months of TCZ treatment [70].

Another multicenter study by Calvo Rìo et al. assessed 25 patients (47 affected eyes) with severe uveitis, complicated by CME in nine cases and several ocular sequalae, including cataracts, glaucoma, synechiae, band keratopathy, maculopathy, and amblyopia. Before TCZ administration, patients had been treated with corticosteroids, conventional immunosuppressive drugs, and biologic agents, including adalimumab, etanercept, infliximab, abatacept, rituximab, anakinra, and golimumab. After 6 months of treatment, 79.2% of patients demonstrated amelioration in anterior chamber inflammation, while 88.2% showed an improvement after 1 year. Other positive outcome of TCZ included central macular thickness reduction in patients with CME and a significant reduction in the prednisone dosage. After a median follow-up of 1 year, visual improvement remained stable, and complete remission of uveitis was noticed in 19 of 25 patients [71]. Other studies over the years have confirmed the efficacy and safety of intravenous TCZ [72,73]. 

Macular edema is one of the major causes of legal blindness in patients with uveitis [72,74,75] and is often present in uveitis, ranging from 20% to 70%, depending on the tests used to assess it (i.e., fundus examination, fluorescein angiography, optical coherence tomography). 

TCZ can be an effective resource when targeting CME. Adan et al. reported one of the first cases of uveitis-related CME refractory to conventional and biologic immunosuppressive therapy, which resolved upon administration of intravenous TCZ. TCZ has proven to be effective not only in treating CME but also in maintaining remission in up to 24 months [72,76,77]. Furthermore, in a study by Masquida and colleagues, TCZ was withdrawn in five patients with sustained remission at month 12, but, in all of them, CME relapsed between 1 and 3 months after TCZ discontinuation. However, when TCZ was reintroduced, it led to uveitis recovery and CME resolution [76]. 

TCZ is also deliverable in a subcutaneous (SC) formulation, and switching from intravenous (IV) to SC administration might ensure flexibility, patient compliance, and better quality of life to patients. A great deal of clinical trials have denoted the safety and efficacy profile of TCZ-SC to be non-inferior to that of TCZ-IV in various rheumatologic conditions [78]. On the back of this promising scenario, it is not surprising that the same approach was attempted in JIA-U. Quesada-Masachs et al. gained a sustained clinical response with TCZ-IV in four children affected by JIA-U. However, when switched to the subcutaneous formulation, all four patients experienced a flare (ocular and/or joint) within a few months [79]. Despite these first findings, other authors have pursued the IV-SC switch. A case of a 6-year-old female patient with treatment-refractory CME who had switched from TCZ-IV to TCZ-SC, because of grade 1 neutropenia development, reported successful control of ocular inflammation [80]. Furthermore, in a just published retrospective study by Marino and colleagues, TCZ-SC has been demonstrated to be efficacious and safe in patients with JIA and uveitis refractory to several bDMARDs [81]. Other studies, such as the APTITUDE trial, were partially in disagreement with these findings. The primary endpoint of the study was the achievement of a two-step decrease or decrease to zero of inflammation in the anterior chamber of the eye, at week 12, in patient refractory to anti-TNF. Twenty-two patients were enrolled, and a phase III trial would have been justified if at least seven had responded. The primary endpoint was not met by TCZ-SC, suggesting that subcutaneous formulation may be less effective in decreasing the level of eye inflammation in U-JIA. However, as the authors themselves claim in the discussion of the study, there were several limitations to the trial itself, especially the tight timing chosen for meeting the primary endpoint, its criteria, and sample heterogeneity [82]. A summary of the aforementioned studies is reported in Table 1.

## 5. Conclusions and Future Perspectives

JIA is the most frequent rheumatic disease in children. JIA-associated uveitis is a major issue because of its sight-threatening complications [17]. Numerous studies and some trials have established short- and long-term efficacy and safety of TCZ in JIA, JIA-U, and in other non-infectious uveitis. 

Since our awareness of the pathogenic mechanisms underlying this disease is poor, it is hard to establish which JIA patients are at risk of developing uveitis and which is the best treatment strategy. Identification of potential risk factors for JIA-associated uveitis appears to be a promising field of research. 

As observed in transcriptomic and proteomic analysis of iris and aqueous humor in patients with JIA-U, B-cells dysregulation leads to IL-6 hyperproduction [25].

This may imply that by inhibiting the activity of overexpressed signaling proteins, such as IL-6 mediated signaling, TCZ has a targeted action on anterior chamber inflammation. On these bases, the use of TCZ is emerging as a valuable drug for the treatment JIA-U, especially treatment-refractory and complicated ones. 

Real life clinical experience in treating patients with JIA-U has shown that clinical response may vary in each patient, and loss of efficacy of a specific treatment occurs during the life-long course of this type of uveitis. Expanding the therapeutic armamentarium of systemic immunosuppressive agents to treat the more aggressive and recalcitrant cases is of paramount importance in the management of JIA-U, one of the most challenging types of uveitis.

IL-6 binds to a membrane-bound IL-6 receptor (IL-6R) or a soluble receptor (sIL-6R) generated by the cleavage of IL-6R by a membrane-bound metalloprotease (ADAM 17). The complex of IL-6 and IL-6R induces the dimerization of a glycoprotein of 130 kDa (gp130) initiating intracellular signaling via the Janus kinase (JAK), signal transducer and activator of transcription (STAT), and rat sarcoma proto-oncogene (RAS) and mitogen-activated protein kinase and phosphoinositide-3 kinase (PI 3K) pathways. All cells express gp130, but only a few cells such as hepatocytes and some leukocytes express IL-6R. Both trans and classic signaling lead to the activation of similar intracellular pathways, but the classic signaling seems to promote anti-inflammatory effects, while IL-6 trans-signaling induces pro-inflammatory responses [83].

## Figures and Tables

**Figure 1 children-10-00434-f001:**
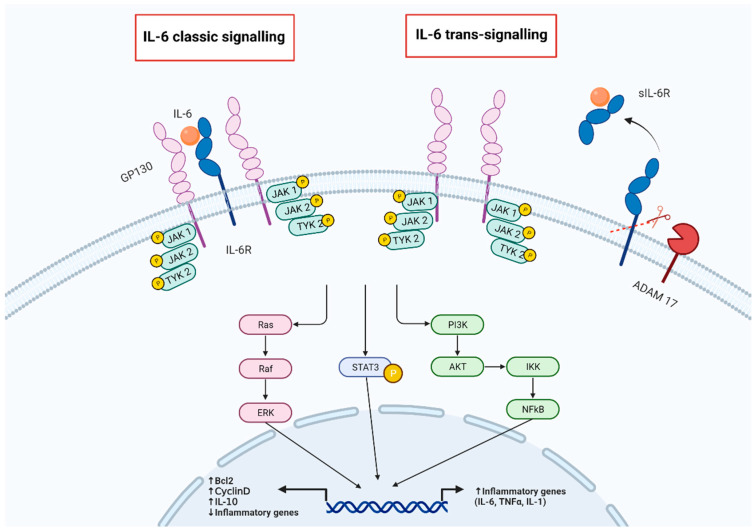
IL-6 pathways.

**Table 1 children-10-00434-t001:** Tocilizumab in U-JIA treatment.

Authors	Number/Sex	JIA	Uveitis Type	Concomitant Use of Steroids	Active Uveitis Assessment	Treatment Prior to TCZ	Ocular Sequelae at Initiation of TCZ	Outcome
Tsang et al. [67]	1, M	Undifferentiated	Anterior	yes	SUN criteria	ADA, IFX, CSA, ETN, ABT	Glaucoma, cataract, macular edema, keratopathy	Inactive
Tappeiner et al. [68]	1, M 2, F	Polyarticular (2), Oligoarticular (1)	Anterior	yes	SUN criteria	PDN, MTX, ETN, ADA, ABT, AZA	Cataract, synechiae, macular edema, glaucoma	Active (1) Inactive (2)
Tappeiner et al. [70]	14, F 3, M	Oligoarticular (9), Polyarticular (7), Undifferentiated (1)	Anterior	yes (14)	SUN criteria	MTX, CSA, MMF, LEF, AZA, INX, ADA, ETN, RTX, golimumab	Cataract (10), band keratopathy (9), synechia (7), ocular hypertension (4), glaucoma (4), macular edema (5), epiretinal membrane (3), optic disc edema (4), ocular hypotony or phthisis (2), retinal detachment (1), amblyopia (1)	Inactive (11) Active (6)
Calvo-Río et al. [72]	21, F 4, M	Oligoarticular (17), Polyarticular (5), Psoriatic JIA (2), ERA (1)	Anterior (17), Panuveitis (4), Intermediate (2), Posterior (2)	yes (11)	SUN criteria	MTX, CSA, LEF, MMF, HCQ, ETN, IFX, ABA, RTX, ABT	Cataract (13), glaucoma (7), synechiae (10), band keratopathy (12), maculopathy (9), amblyopia (5)	Active (6) Inactive (11)
Maleki et al. [73]	8	Polyarticular (5), Oligoarticular (3)	Anterior (14), Intermediate Uveitis (4), Retinal vasculitis (9), Papillitis (4)	unknown	SUN criteria	MTX, IFX, ADA, AZA, ABT, ETN, MMF, IVIg, RTX, Certolizumab	Cataract (4)	Active (3) Inactive (5)
Mesquida M. et al. [77]	7, F	Unknown	Unknown	yes (7)	SUN criteria	PDN, MTX, CSA, ADA, RTX, IFX, ABT	Unknown	Inactive (7)
Adán A. et al. [80]	1, F	Oligoarticular	Macular edema	unknown	Not defined (Visual acuity?)	PDN, MTX, IFX, ADA	Glaucoma	Inactive (1)
Marino A. et al. [81]	9, F 4, M	Oligoarticular (9), Polyarticular (4)	Anterior (8), Panuveitis (5)	yes	SUN criteria	PDN, MTX, CSA, ADA, IFX, RTX	Band keratopathy, cataract, posterior synechia	Inactive (5) One-step improvement (1)

ABT = abatacept, PDN = prednisone, MTX = methotrexate, ADA = adalimumab, ETN = etanercept, IFX = infliximab, RTX = rituximab, IVIg = endovenous immunoglobulins, CSA = cyclosporine A, MMF = mycophenolate mofetil, AZA = azathioprine, HCQ = hydroxychloroquine, LEF = leflunomide, SUN = Standardization of Uveitis Nomenclature.

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
