# Peer review of "Tocilizumab in Juvenile Idiopathic Arthritis Associated Uveitis, a Narrative Review"

_children, 2023, doi:10.3390/children10030434_

Round 1

Reviewer 1 Report

The review provides a concise overview of JIA associated uveitis, its treatment and the potential utility of treatment with tociluzimab. It will be a useful source for the area. A few of the cited references appear to be from the last author (E. Miserocchi). Please specify in the review text if the references with E. Miserocchi as first or last author refers to the last author of the submitted review.

 A few other clarifications are needed, please see below.

Line 13: ‘faint’ – could you please replace with a more precise definition? I am not sure what exactly the authors mean.

Line 60: please define ACR

Line 177: 100% of patients – please provide the number of patients

Lines 193-197: Please clarify – is the JUVE-BRIGHT trial already concluded or still in progress?

Line 233: the authors use ‘mIL-6R’ in this sentence but not in others. Please define ‘mIL-6R’ and check for a consistent nomenclature for the IL6 receptor.

Line 243: instead of ‘little’ please use ‘young’

Author Response

Dear reviewer. Thank you for your comments.

Concerning your first point, yes E. Miserocchi cited in the review is also the last author of this paper as you may see from the references. In one specific sentence I have replaced it with "our group". 

We have proceeded to change and specify what you asked for. 

2. faint replaced by difficult to detect

3. ACR = American college of rheumatology

4. 7 patients

5. JUVE bright is still ongoing. 

6. We have done it

7. We have done it

Thanks. 

Reviewer 2 Report

This is an interesting overview, but the title topic starts at page 6.JUVE-BRIGHT  trial – is running – no data published

The most current therapeutic recommendations of the MIWGUC are not mentioned

In the part tocilizumab in non infectios uveitis- the effect of tocilizumab is shown for disease, which are not related to JIA and does not belong to the topic

Reference 70 refers to sc Tocilizumab in RA, not for JIA uveitis- this extrapolation need to be discussed

It would be great to have a table of the cases/case series treated with Tocilizumab in JIA associated uveitis, with patient caractericitic, how effect was assessed, when effect occured,...

The phase II study of Toicilizumab is very shortly discussed

Author Response

I am sorry, I have found a way. Here is the table

Thank you

Table 1. Tocilizumab in U-JIA treatment

Authors

Number/sex

JIA

Uveitis type

Concomitant use of steroids

Active uveitis assessment

Treatment prior to TCZ

Ocular sequelae at initiation of TCZ

Outcome

Tsang et al. 59

1, M

Undifferentiated

Anterior

yes

SUN criteria

ADA, IFX, CSA, ETN, ABT

Glaucoma, cataract, macular edema, keratopathy

Inactive

Tappeiner et al. 60

1, M

2, F

Polyarticular (2)

Oligoarticular (1)

Anterior

yes

SUN criteria

PDN, MTX, ETN, ADA, ABT, AZA

Cataract, synechiae, macular edema, glaucoma

Active (1)

Inactive (2)

Tappeiner et al. 62

14, F

3, M

Oligoarticular (9)

Polyarticular (7)

Undifferentiated (1)

Anterior

yes (14)

SUN criteria

MTX, CSA, MMF, LEF, AZA, INX, ADA, ETN, RTX, golimumab

Cataract (10),

band keratopathy (9),

synechia (7),

ocular hypertension (4),

glaucoma (4), macular edema (5) piretinal membrane (3)

Optic disc edema (4)

Ocular hypotony or phthisis (2)

Retinal detachment (1)

Amblyopia (1)

Inactive (11)

Active (6)

Calvo-Río et al. 63

21, F

4, M

Oligoarticular (17)

Polyarticular (5)

Psoriatic JIA (2)

ERA (1)

Anterior (17)

Panuveitis (4)

Intermediate (2)

Posterior (2)

yes (11)

SUN criteria

MTX, CSA, LEF, MMF, HCQ, ETN, IFX, ABA, RTX, ABT

Cataracts (13)

Glaucoma (7)

Synechiae (10)

Band keratopathy (12)

Maculopathy (9)

Amblyopia (5)

Active (6)

Inactive (11)

Maleki et al. 65

8

Polyarticular (5)

Oligoarticular (3)

Anterior (14)

Intermediate Uveitis (4)

Retinal vasculitis (9)

Papillitis (4)

unknown

SUN criteria

MTX, IFX, ADA, AZA, ABT, ETN, MMF, IVIg, RTX, Certolizumab

Cataracts (4)

Active (3)

Inactive (5)

Mesquida M. et al.69

7, F

Unknown

Unknown

yes (7)

SUN criteria

PDN, MTX, CSA, ADA, RTX, IFX, ABT

Unknown

Inactive (7)

Adán A. et al. 72

1, F

Oligoarticular

Macular edema

unknown

Not defined

(Visual acuity?)

PDN, MTX, IFX, ADA

Glaucoma

Inactive (1)

Marino A. et al.73

9, F

4, M

Oligoarticular (9)

Polyarticular (4)

Anterior (8)

Panuveitis (5)

yes

SUN criteria

PDN, MTX, CSA, ADA, IFX, RTX

Band keratopathy, Cataract

Posterior synechia

Inactice (5)

One-step improvement (1)

ABT= abatacept, PDN=prednisone, MTX=methotrexate, ADA=adalimumab, ETN=etanercept, IFX=infliximab, RTX=rituximab, IVIg= endovenous immunoglobulins, CSA=cyclosporine A, MMF=mycophenolate mofetil, AZA=azathioprine, HCQ=hydroxychloroquine, LEF=leflunomide

SUN = Standardization of Uveitis Nomenclature

Reviewer 3 Report

Journal   children

Article Tocilizumab in Juvenile Idiopathic Arthritis associated Uveitis, a narrative review.

This manuscript covers a hot topic. The authors talked about the role of tocilizumab in detail. The manuscript is well written .However ,some points need to be addressed

29-32…many sentences without references

35.. autoimmune characteristics (ANA and RF positivity) …this sentence is to some extent misleading as ANA is not a parameter to classify JIA

46-50…again, a lot of knowledge without references

63…there is a need to add references in their proper sites.

It would be better to add a figure to illustrate the role of il6 antagonist in uveitis.

It will be better if the section pathogenesis of uveitis clearer.

The section of treatment of jia is well written ,however the authors did not refer to corticosteroids and their role

4. Tocilizumab

Considering tocilizumab as a line of treatment, so ,it is unlogic to add tocilizumab as a separate title from treatment .It will be better to subtitle the treatment section

213.. plasmacells…. English editing and grammar revision are mandatory.

239.. diseases 51,52].……a bracket is missing.

The titles are in need to be rearranged and some titles need to be subtitles

Author Response

Dear reviewer. Thanks for your comments. I cannot add the references directly in the text so I have reported them here. I meanwhile asked the editor to add them to the original paper. 

1.

  • Juvenile Idiopathic Arthritis: Diagnosis and Treatment 10.1007/s40744-016-0040-4
  • International League of Associations for Rheumatology classification of juvenile idiopathic arthritis: second revision, Edmonton, 2001 PMID: 4760812
  • Incidence and outcomes of uveitis in juvenile rheumatoid arthritis, a synthesis of the literature https://doi.org/10.1007/s00417-005-0087-3
  • Risk markers of juvenile idiopathic arthritis-associated uveitis in the Childhood Arthritis and Rheumatology Research Alliance (CARRA) Registry 3899/jrheum.130302

2. We have corrected it. It is highlighted in blue. RF positivity is part of the classification.

3.

  • Incidence and outcomes of uveitis in juvenile rheumatoid arthritis, a synthesis of the literature https://doi.org/10.1007/s00417-005-0087-3
  • Risk markers of juvenile idiopathic arthritis-associated uveitis in the Childhood Arthritis and Rheumatology Research Alliance (CARRA) Registry 3899/jrheum.130302
  • Prevalence of uveitis in an outpatient juvenile arthritis clinic: onset of uveitis more than a decade after onset of arthritis. J Pediatr Ophthalmol Strabismus doi: 10.3928/0191-3913-19970301-09.

4. The sentence in line 63 sentence is introductory to all the articles we have included in the review. 

5. In paragraph 3 we have discussed the use corticosteroids in U-JIA. The use of steroids in JIA is beyond the scope of this review. 

6. Considering tocilizumab is the focus of the review we have thought to describe its mechanism of action in details and giving it  a separate paragraph.We have rearranged the paragraphs, but we are open to suggestions.

7. plasma cells has been corrected (highlighted in blue).

8. Thank, you we have corrected.

9. We have rearranged the paragraphs and add some subtitles. We are open to further suggestions.

As regard as the figure, since the pathogenesis of uveitis is so unclear it is difficult to produce such image  but we have tried to exemplify it in the graphical abstract we have submitted. 

Thank you

Round 2

Reviewer 3 Report

The manuscript has been improved